# CuS/rGO-PEG Nanocomposites for Photothermal Bonding of PMMA-Based Plastic Lab-on-a-Chip

**DOI:** 10.3390/nano11010176

**Published:** 2021-01-12

**Authors:** Young Jae Kim, Jae Hyun Lim, Jong Min Lee, Ji Wook Choi, Hyung Woo Choi, Won Ho Seo, Kyoung G. Lee, Seok Jae Lee, Bong Geun Chung

**Affiliations:** 1Department of Mechanical Engineering, Sogang University, Seoul 04107, Korea; kootng@naver.com (Y.J.K.); delete1821@gmail.com (J.W.C.); gmpwoo@naver.com (H.W.C.); 2Department of Biomedical Engineering, Sogang University, Seoul 04107, Korea; boy6635@naver.com (J.H.L.); d778503@naver.com (W.H.S.); 3Division of Chemical Industry, Yeungnam University College, Daegu 38541, Korea; caboera85@naver.com; 4Division of Nano-Bio Sensors/Chips Development, National NanoFab Center, Daejeon 34141, Korea; kglee@nnfc.re.kr (K.G.L.); sjlee@nnfc.re.kr (S.J.L.)

**Keywords:** photothermal bonding, PMMA-based plastic lab-on-a-chip, CuS/rGO-PEG nanocomposite

## Abstract

We developed copper sulfide (CuS)/reduced graphene oxide (rGO)-poly (ethylene glycol) (PEG) nanocomposites for photothermal bonding of a polymethyl methacrylate (PMMA)-based plastic lab-on-a-chip. The noncontact photothermal bonding of PMMA-based plastic labs-on-chip plays an important role in improving the stability and adhesion at a high-temperature as well as minimizing the solution leakage from microchannels when connecting two microfluidic devices. The CuS/rGO-PEG nanocomposites were used to bond a PMMA-based plastic lab-on-a-chip in a short time with a high photothermal effect by a near-infrared (NIR) laser irradiation. After the thermal bonding process, a gap was not generated in the PMMA-based plastic lab-on-a-chip due to the low viscosity and density of the CuS/rGO-PEG nanocomposites. We also evaluated the physical and mechanical properties after the thermal bonding process, showing that there was no solution leakage in PMMA-based plastic lab-on-a-chip during polymerase chain reaction (PCR) thermal cycles. Therefore, the CuS/rGO-PEG nanocomposite could be a potentially useful nanomaterial for non-contact photothermal bonding between the interfaces of plastic module lab-on-a-chip.

## 1. Introduction

Lab-on-a-chip has great potential in various research fields, such as biological analysis, chemical mixing, and synthesis. To perform high-throughput screening and various functions within a lab-on-a-chip, the design of a lab-on-a-chip may be complicated. To address this limitation, the concept of the plastic module chip that can easily combine and bond several lab-on-a-chip is required [1]. The bonding technology is of great importance to determine the physical and mechanical properties of the microfluidic chip-to-chip interfaces. The chip bonding can improve the stability (e.g., thermal shock resistance [2]), prevent the adhesion reduction due to moisture penetration into the adhesion interface [3], and minimize the gap between chip-to-chip interfaces [4]. Poly (dimethylsiloxane) (PDMS) and poly (methyl methacrylate) (PMMA) have widely been used in a microfluidic device for chemical and biological applications [5,6]. Although the PDMS-based microfluidic device can be easily bonded [7], it still has limitations, such as the requirement of an external oxygen plasma treatment equipment, its ease to be deformed by external stress, and evaporation of the sample in a high-temperature [8]. Due to these limitations, PDMS is not a suitable material for mass production and commercialized labs-on-chip. In contrast, PMMA is a representative isotropic material with high optical transmittance and low refractive index [9]. We can achieve mass production of the PMMA-based plastic chip, because it can be easily produced in the desired pattern through injection molding, resulting in a lower cost of the final plastic chip. Thus, PMMA is known as one of the most suitable materials for commercialization, because it is cost-effective as compared to glass or other silicon materials. Additionally, PMMA has stable mechanical properties that can prevent evaporation of samples even at a high temperature. However, the use of adhesives can create gaps between PMMA plastic labs-on-chip, which can lead to leaks [10], and the toxicity of the adhesives can adversely affect the biological samples [11]. To address these problems, the biocompatible nanocomposite can be used as an alternative bonding adhesive.

A number of technologies have previously been studied for bonding of the plastic lab-on-a-chip, such as adhesive-based bonding, solvent-based bonding, and thermal bonding [12]. In the case of the conventional adhesive-based bonding, there was the gap problem in the bonding surface due to the thickness of the bonding layer and this gap could cause leakage [10,13]. The conventional solvent-based bonding has used various solvents (e.g., acetone, dichloromethane, methyl ethyl ketone, and chloroform) for high adhesion [14]. However, the conventional solvent-based bonding showed some limitations in that the channel deformation was generated by toxic solvents, and it was difficult to locally attach to the plastic lab-on-a-chip [13,15]. The thermal bonding has widely been employed for PMMA-based plastic labs-on-chip. Despite the strong adhesion force of thermal bonding, the heat-induced channel distortion is still challenging [16,17,18]. For the biological applications of the plastic lab-on-a-chip, the stability and applicability of the bonding method needed to be considered [19]. As a method of securing the geometrical stability of the microchannel, conventional laser-based bonding has been used [20]. However, its process was expensive [21,22].

To overcome the limitations of conventional bonding methods, plastic labs-on-chip can be bonded using the photothermal effect of nanomaterials. The nanomaterial-based bonding method uses the plasmonic phenomenon in which the nanomaterial can convert the light into heat energy. It can minimize damage to biological samples, because it generates the heat in a specific area by near-infrared (NIR) laser. Among photothermal nanomaterials, reduced graphene oxide (rGO) has widely been used in various research fields due to unique physiochemical properties [23]. However, since rGO has low solubility in aqueous solvents and has hydrophobic properties, its biological applications are limited, and it is also difficult to use for bonding to a hydrophobic substrate. Therefore, the previous studies have been conducted to modify the hydrophobic rGO surface to be hydrophilic [24]. In previous studies, the pristine rGO was coated with a hydrophilic polymer, such as poly (ethylene glycol) (PEG), and showed that PEG improved the hydrophilicity of graphene oxide (GO). Furthermore, rGO can be used to significantly increase the temperature in combination with other photothermal nanomaterials. Copper sulfide (CuS) nanoparticles can be used as a photothermal agent due to their inherent absorption in the NIR wavelength region [25]. Thus, the combination of rGO and CuS can be used to rapidly increase the temperature via a NIR laser irradiation [26]. In this paper, we developed the novel non-contact-based photothermal bonding technology in PMMA-based plastic labs-on-chip using CuS/rGO-PEG photothermal nanocomposites.

## 2. Materials and Methods

### 2.1. Fabrication of PMMA-Based Plastic Lab-on-a-Chip

The plastic lab-on-a-chip was fabricated from a PMMA plate with dimensions of 100 mm width, 100 mm length, and 1 mm depth on a micromilling machine (T60, Tinyrbo, Incheon, Korea). This machine was equipped with a spindle motor and 1 mm-diameter mill. To construct the microchannel on the plate, the pristine PMMA plate was loaded on the holder and the end mill was rotated with a 20,000 rpm to transfer the channel design. The spindle attached motor heads were controlled by a computer using G-code programming (Visual Mill, MecSoft Corporation, Irvine, CA, USA). Therefore, the plastic lab-on-a-chip with dimensions of 40 mm width, 40 mm length, and 0.1 mm channel depth was fabricated.

### 2.2. Synthesis of CuS/rGO-PEG Nanocomposites

Copper chloride (CuCl_2_), sodium sulfide (Na_2_S), citric acid, hydrazine monohydrate, N-(3-Dimethylaminopropyl)-N′-ethylcarbodiimide hydrochloride (EDC), and N-hydroxysuccinimide (NHS) were purchased from Sigma-Aldrich (St. Louis, MO, USA). Amino PEG (mPEG-NH_2_) was obtained from NANOCS, Inc. (New York, NY, USA). GO aqueous solution was purchased from Graphene square, Inc. (Seoul, Korea). rGO-PEG nanocomposites were prepared according to the previously reported methods [24]. One hundred milligrams of EDC and NHS were added to a GO solution (1 mg/mL) and were ultrasonicated. After 1 h, 100 mg of PEG-NH_2_ was stirred for 18 h. To remove unacted chemicals, it was dialyzed with a cellulose membrane (MWCO 6–8 kDa, Spectrum Laboratories, Inc., New Brunswick, NJ, USA) for two days. The obtained material was lyophilized for two days to make dried PEG-GO nanocomposites. To reduce GO-PEG nanocomposites, GO-PEG was stirred with 0.05% of hydrazine monohydrate for 30 min at 90 °C. The rGO-PEG was purified by a dialysis membrane (MWCO 3500, Spectrum Laboratories, Inc., New Brunswick, NJ, USA) against deionized water (DI water) for two days and the final product was freeze-dried for two days. CuS nanoparticles were also prepared according to the previously reported methods [27]. Briefly, 14 mg of CuCl_2_ and 20 mg of citric acid were dissolved in 100 mL DI water. After stirring for 30 min, 7.8 mg of Na_2_S was added and was subsequently stirred for 15 min at 90 °C. Finally, CuS nanoparticles were freeze-dried for two days.

### 2.3. Characterization of CuS/rGO-PEG Nanocomposites

The morphology and size of CuS, rGO-PEG, and CuS/rGO-PEG were observed by transmission electron microscopy (TEM, JEOL−2100F, JEOL Ltd., Tokyo, Japan). To confirm the chemical bonding and surface modification, GO-PEG and rGO-PEG were analyzed using Fourier transform infrared spectroscopy (FT-IR, Nicolet 6700, Thermo Scientific Inc., Waltham, MA, USA) using KBr pellets ranging from 400 to 4000 cm^−1^. The surface charge of GO, GO-PEG, and rGO-PEG was measured by using a Zetasizer Nano Z (Malvern Instruments, Malvern, UK). The optical properties of CuS, GO-PEG, rGO-PEG, and CuS/rGO-PEG were measured by UV-vis spectroscopy (UV 1800, Shimazu, Kyoto, Japan). To investigate the photothermal effect, dispersed rGO-PEG aqueous solution (0.25 mg/mL) with CuS (0−1.0 mg/mL) was irradiated for 10 min using an 808 nm NIR laser (MDL-N−808, CNI Optoelectronics Tech. Co. Ltd., Changchun, China) at a laser intensity of 1 W/cm^2^. The change of temperature was recorded by a thermosensor (DTM−318, Tecpel Co., New Taipei City, Taiwan) every 1 min. In addition, the CuS/rGO-PEG nanocomposite solution was dropped on the PMMA-based plastic lab-on-a-chip. Subsequently, the NIR laser was irradiated and the temperature was monitored using an infrared thermal imaging camera (E60, FLIR Inc., Wilsonville, OR, USA).

### 2.4. Experimental Setup for Photothermal Bonding

To improve the adhesion between PMMA-based plastic lab-on-a-chip, the wettability of the CuS/rGO-PEG nanocomposites was firstly removed through a preheating process and was carefully dropped on the surface of the second, bottom PMMA layer plate before stacking the PMMA-based plastic lab-on-a-chip (Figure 1A). We used a 2.5 W/cm^2^ NIR laser intensity and the preheating was performed for 5 s per drop on the surface of the bottom second PMMA layer plate. When the preheated CuS/rGO-PEG nanocomposites on the bottom second PMMA layer plate was stacked to the first, top PMMA layer plate, the secondary NIR laser was subsequently irradiated for 10 s to complete photothermal bonding.

### 2.5. Analysis of Mechanical Properties for Photothermal Bonding

To analyze the mechanical properties of the photothermal bonded PMMA-based plastic lab-on-a-chip, the tensile force and gap length of the bonded PMMA-based plastic lab-on-a-chip were measured. Various volumes (1, 2, 3, 4, and 5 μL) of CuS/rGO-PEG nanocomposites were dropped onto the adhesion site. The photothermal bonding between the first and second PMMA layer plates was performed using NIR laser irradiation. The tensile force of the bonded PMMA-based plastic lab-on-a-chip was measured with a tensile compressor (Mecmesin, MultiTest2.5-I, Slingford, UK) and the data was analyzed until the photothermal bonding was broken by pulling both sides of the PMMA plate at a rate of 2 mm/min. In addition, the tensile force with respect to the number of drops (one to five drops) was analyzed. To confirm the leakage after bonding by the CuS/rGO-PEG nanocomposites, the gap length of the adhesion site was measured with respect to the number of drops of the CuS/rGO-PEG nanocomposite. The adhered area was observed with an inverted fluorescent microscope (Olympus, Tokyo, Japan) and the gap length was analyzed using Image J (National Institute of Health, Bethesda, MD, USA) software. The various fluids (DI water, ethyl alcohol, 94.5% (DAEJUNG, Siheung, Korea), dimethyl sulfoxide, 99.8% (SAMCHUN, Seoul, Korea) and mineral oil (Sigma Aldrich, St. Louis, MO, USA)) were applied to the bonded PMMA-based plastic lab-on-a-chip. The flow rate of fluids was increased by 1 μL/min (DI water, dimethyl sulfoxide) and 10 μL/min (mineral oil) every 10 s, to confirm leakage from the plastic microfluidic channel.

### 2.6. Polymerase Chain Reaction (PCR) Applications

PCR experiments were performed according to the manufacturer’s instruction (Applied Biosystems, Foster City, CA, USA). For PCR experiments, 80 µL of the PCR mixture solution consisted of 45 µL of a commercial master mix (Quant Studio 3D Digital PCR Master Mix V2, Applied Biosystems, Foster City, CA, USA), 9 µL of forward primer (5′-TGGTGCTGGTTCTGATAAAGGAG−3′), 9 µL of reverse primer (5′-GAATCTGCATCAGAGACAAAGTCA−3′), 9 µL of FAM probe, and 8 µL of cDNA template (final concentration: 104 copies/µL). After photothermal bonding between PMMA-based plastic lab-on-a-chip, 20 µL of the PCR solution was pipetted into the PMMA-based plastic lab-on-a-chip and the inlet was sealed with a self-adhesive PCR sealing tape (4titude, Dorking, UK) to prevent water evaporation. The PMMA-based plastic lab-on-a-chip was then placed onto a thermocycler (DH200, BioD, Gwangmyeong, Korea) and PCR was conducted with 35 cycles of denaturation at 95 °C for 60 s, annealing at 55 °C for 60 s, and extension at 72 °C for 60 s. To analyze the fluorescence intensity, the PMMA-based plastic lab-on-a-chip was imaged using an inverted fluorescence microscope and the images were analyzed using Image J software.

### 2.7. Statisitical Analysis

The statistical analysis was performed by a Student’s *t*-test with ** *p* < 0.01 considered statistically significant.

## 3. Results and Discussion

### 3.1. Fabrication of PMMA-Based Plastic Lab-on-a-Chip and Analysis CuS/rGO-PEG Nanocomposites

We fabricated the plastic lab-on-a-chip consisting of two PMMA layer plates (Figure 1A). The PMMA-based plastic lab-on-a-chip has great advantages as compared to conventional PDMS-based chips, such as it can prevent the evaporation of biological samples at a high temperature, provide mass production, and inhibit deformation. In the manufacturing process, a three-dimensional (3D) drawing was designed by computer software, and it converted the designed file to the micromilling machine. The micromilling machine was then operated to fabricate the desired micropatterns on the PMMA plastic plate. The CuS/rGO-PEG nanocomposites solution was carefully patterned on the second, bottom PMMA layer plate. Afterwards, the NIR laser-mediated preheating removed the wettability of the CuS/rGO-PEG nanocomposite solution. The first, top PMMA layer plate was aligned to the second, bottom PMMA layer plate patterned with the CuS/rGO-PEG nanocomposites. The first PMMA layer plate was composed of one inlet, one outlet, and the microchannel, which was connected to the inlet. The second PMMA layer plate only contained the microchannel which could be connected to the outlet after alignment. The PMMA-based plastic lab-on-a-chip were combined using the photothermal bonding. For the photothermal bonding, we prepared the mixture of CuS/rGO-PEG nanocomposites (Figure 1C) and their physicochemical properties were characterized by using analytical equipment. 

The use of rGO still suffers for various biological applications due to poor colloidal stability in an aqueous solution [28]. To solve this problem, we functionalized PEG on the surface of GO to overcome limited solubility and subsequently mixed with CuS nanoparticles to enhance the photothermal effect. The size and morphology of CuS, rGO-PEG, and CuS/rGO-PEG were analyzed by TEM (Figure 2A). The sphere-types of CuS nanoparticles were uniformly synthesized and the average diameters were observed about 10 nm, which was similar to previous works [29]. The rGO-PEG showed single-layer sheets within 1 μm. Interestingly, in this case of CuS/rGO-PEG nanocomposites, a number of CuS nanoparticles were covered on the surface of the single-layer rGO-PEG sheet. It revealed that CuS/rGO-PEG nanocomposites were synthesized by simple physical mixing [30]. Additionally, we observed TEM images with different ratios of CuS to rGO-PEG (1:1 and 2:1), showing that it did not find any morphology change with respect to CuS ratio (data not shown). 

To confirm the chemical conjugation of amino PEG on single-layer GO sheets, GO and rGO-PEG were measured by FT-IR spectroscopy (Figure 2B). The pristine GO showed the inherent stretching vibration peaks at 1720 cm^−1^ and 3390 cm^−1^, corresponding to carboxyl and hydroxyl group, respectively [31]. After conjugation of amino PEG on rGO nanosheets, the new peaks were observed at 1466 cm^−1^ and 1340 cm^−1^, indicating –CH_2_ and CH_3_ stretching in PEG. In addition, since amino PEG was anchored to GO via EDC-NHS coupling reaction, amide I and II vibration peaks were observed at 1640 cm^−1^ and 1455 cm^−1^ [32]. Furthermore, the broad band around 3420 cm^−1^ corresponded to O-H stretching vibrations and this absorption was diminished in rGO-PEG, indicating that the reduction process was successfully performed [24]. The zeta potential measurements were conducted to further verify the formation of rGO-PEG in Figure 2C. The zeta potential of GO-PEG showed a negative surface charge (−27.5 mV) and it was higher than that of pure GO (−37.5 mV). After the reduction process (rGO-PEG), the zeta potential increased from −27.5 mV to −17.2 mV. These changes were attributed to the conjugated PEG and reduction, showing successful synthesis of rGO-PEG [18,33]. To evaluate an optical property, CuS, rGO-PEG, and CuS/rGO-PEG nanocomposites were measured by UV-vis spectroscopy (Figure 2D). The CuS nanoparticle was one of the famous photothermal agents and broad absorption peaks appeared strongly in the NIR range (700 to 800 nm) [27]. rGO-PEG has also displayed an absorption band at the same range, but its intensity was not higher than that of CuS nanoparticles. Interestingly, in the NIR wavelength range, the CuS/rGO-PEG nanocomposites exhibited a much enhanced absorption peak, which might have been caused by the mixture of CuS and rGO-PEG [34]. Next, we investigated the photothermal property of CuS/rGO-PEG nanocomposites in Figure 2E. Various mixture solutions based on CuS/rGO-PEG nanocomposites were exposed to 808 nm NIR laser irradiation at a density of 1 W/cm^2^ for 10 min. As a control, rGO-PEG (0.25 mg/mL) was performed under the same condition and temperature of rGO-PEG increased to 10.5 °C. To determine the optimized condition of nanocomposites, we decided to use the rGO-PEG solution (0.25 mg) and CuS nanoparticles were subsequently added to the rGO-PEG solution at a weight ratio of one to four. As the amount of CuS nanoparticles was increased, the temperature of CuS/rGO-PEG nanocomposites elevated to 15.2, 18.0, 21.3, and 21.9 °C, respectively. However, when the amount of CuS nanoparticles was added more than three times, the difference in temperature rise of the CuS/rGO-PEG nanocomposite was not noticeable. We carefully speculated that the temperature did not rise anymore, because the CuS nanoparticles were completely covered on the rGO-PEG surface. Therefore, CuS/rGO-PEG nanocomposite has been optimized (weight ratio 3:1) and can be used for noncontact photothermal bonding experiments. 

We confirmed that the temperature increased when the NIR laser irradiated the area where the CuS/rGO-PEG nanocomposite was placed (Figure 3A). Because nanocomposites exist in a solvent and have wettability, they may interfere with raising the temperature required for bonding. To increase the efficiency of bonding, after a preheating process, the top plate was covered and a laser was irradiated for adhesion. To increase the efficiency of bonding, the solvent was evaporated through a preheating process, and the top plate was covered and a laser was then irradiated for adhesion. In the heating process, it was confirmed that the temperature reached 150 °C, which was sufficient to melt the PMMA plate. In the case of conventional laser bonding, the material is melted and bonded by raising the temperature. In contrast, in the case of photothermal bonding, since heat can be applied only to a specific area where the nanocomposite is located, it is possible to bond to a local area. To optimize the laser intensity, the temperature-time change was analyzed, and the arrival time was measured based on the 150 °C melting point of PMMA (Figure 3B). Though 1 W and 1.5 W of power did not reach 150 °C for 1 min, it was confirmed that 2.5 W or more intensity reached 150 °C within 30 s. In addition, the heating rate per second was calculated while increasing the intensity of the laser (Figure 3C). From the 2.5 W condition, the heating rate did not increase significantly at a specific value of 5 °C/s. In addition, since the energy received by the nanocomposite is large at a higher laser intensity, it may damage the sample. Thus, the laser intensity was optimized to 2.5 W.

### 3.2. Analysis of Mechanical Properties of PMMA-Based Plastic Lab-on-a-Chip

We analyzed the mechanical properties of the PMMA-based plastic lab-on-a-chip bonded by CuS/rGO-PEG nanocomposites. The stress and strain analysis was performed according to the previously reported method [35]. First, the tensile force with respect to the volume and drop number of rGO-PEG/CuS nanocomposites on the PMMA surface was measured with a tensile compressor. The NIR laser intensity (2.5 W/cm^2^) was used for photothermal bonding of the PMMA-based plastic lab-on-a-chip. The stress and strain were analyzed from displacement of the bonded PMMA-based plastic lab-on-a-chip. When the volume of the nanocomposite was changed from 1μL to 5 μL, the smallest volume (1 μL) showed small maximum stress (3.4 MPa) and strain (0.036 ε), resulting in smaller adhesion force (Figure 4A). In the case of 2 μL and 3 μL volume, the stress and strain values were proportional to the volume, indicating the stress values of 5.8 MPa and 6.4 MPa as well as strain values of 0.051 ε and 0.058 ε, respectively. In contrast, in the case of 4 and 5 μL volume, it showed the strain value of 0.046 ε and 0.05 ε as well as a stress value of 6.0 MPa and 6.2 MPa, respectively. The stress values were reduced as compared to the maximum stress value (6.4 MPa) of 3 μL. It was probably due to the moisture on the PMMA plastic surface and this moisture was caused by the wettability increase of the CuS/rGO-PEG nanocomposite solution as the volume was increased. As the wettability was increased, the adhesive energy might be decreased and the adhesive strength could be different according to the volume, as previously described [36]. As a result, we optimized the droplet volume as 3 μL, which had the maximum strain value of 0.058 ε and stress value of 6.4 MPa. Afterward, the stress and strain of the PMMA-based plastic lab-on-a-chip with respect to the number of drops were further analyzed by using a 3 μL volume of CuS/rGO-PEG nanocomposites. The tensile force and displacement of the bonded PMMA-based plastic lab-on-a-chip were also measured with various droplet numbers (one to five drops) (Figure 4B). The stress-strain curve of the bonded PMMA-based plastic lab-on-a-chip was analyzed using Young’s modulus. In the case of one drop, the value of Young’s modulus was 149.73 MPa. It was continuously increased from two to five drops and Young’s modulus values were 149.78 MPa, 149.87 MPa, 149.90 MPa, and 149.92 MPa, respectively.

We confirmed that the adhesion force was also proportional to the droplet number of the CuS/rGO-PEG nanocomposite solution. While increasing the number of drops, the stress and strain increased. However, it needs to be confirmed whether there is any thermal distortion of the PMMA-based plastic lab-on-a-chip, because the fluid leakage may occur through the gap between the adhered surfaces when thermal deformation generates on the PMMA surface. To confirm the leakage at the interface of the plastic lab-on-a-chip, the PMMA-based plastic lab-on-a-chip was bonded with one drop to three drops of CuS/rGO-PEG nanocomposite solution (3 μL volume) and the length of the gap was observed with an inverted microscope. It showed that the gap was 16.9 μm at one drop and was decreased to 12.3 μm at two drops. Similarly, the gap was reduced to 7.65 μm at three drops. As the droplet number of CuS/rGO-PEG nanocomposites was increased, the gap length between the first and second PMMA layer plates was narrowed, because the adhesion area was increased (Figure 4D). The gap length was measured by the gap image taken by a microscope and data were obtained through five different experiments. For the statistical analysis, the *p* value was analyzed by a Student’s t-test, indicating that each *p* value from one drop to three drops was less than 0.01 (** *p* < 0.01). Above three drops, the gap did not show the significant difference.

We also analyzed the leakage with various fluids (e.g., DI water, dimethyl sulfoxide (DMSO), and mineral oil) with different viscosities in the PMMA-based plastic lab-on-a-chip (Figure 5A). Leakage was investigated at the inlet, intersection, and outlet of the bonded PMMA-based plastic lab-on-a-chip while the flow rate was increased every 10 s. In the case of water and DMSO, the leakage was confirmed by increasing 10 μL/min every 10 s from 10 μL/min, and in the case of mineral oil with a significantly high viscosity, the linkage was confirmed by increasing 1 μL/min every 10 s from 1 μL/min. At the initial flow rate, there were no leakages at the inlet and outlet of the bonded PMMA-based plastic lab-on-a-chip under all fluid conditions. In addition, we observed no leakage at the intersection between the first and second PMMA layer plates. To obtain the minimum pressure of the flow rate that the plastic lab-on-a-chip could withstand, the flow rate at the point where the leakage occurred was converted into the pressure. The pressure can be calculated using Poiseuille’s law:$$Q=\;\frac{\pi D_{eff}^{4}\Delta P}{128L\mu }$$
where *Q* is flow rate in $m^{3}s^{-1}$, $D_{eff}$ is the effective diameter of the microchannel, Δ*P* is pressure drop, *L* is the length of the channel and *μ* is viscosity of fluid, as previously described [37]. We monitored the flow rate of fluids until leaking of the bonded lab-on-a-chip occurred on the channel. In the case of water, it was measured at up to 200 μL/min in consideration of the range of flow rate. However, it did not reach the pressure at which water leakage occurred [38]. By the calculation using Poiseuille’s law, the maximum pressure that water flows in the bonded plastic lab-on-a-chip is 10.62 kPa. Similarly, the ethanol viscosity is not significantly different from water. There is no leakage in the range of 200 μL/min and the pressure calculated by Poiseuille’s law is 13 kPa. In addition, we observed that the leakage of DMSO and mineral oil was generated at 23 kPa which occurred at 25 μL/min in oil and 190 μL/min in DMSO (Figure 5B). Since the viscosities of tested fluid types were different, the leakage flow rate range showed a different scale. As a result, we confirmed that the leakage of the bonded PMMA-based plastic lab-on-a-chip was not observed until 22.8 kPa (Figure 5C). 

### 3.3. PCR Applications in the PMMA Plastic Lab-on-a-Chip

To verify the thermal stability and applicability of the bonded PMMA-based plastic lab-on-a-chip, we performed the PCR process with 35 thermal cycles in three temperature ranges (65−95 °C) (Figure 6). After 35 thermal cycles, we analyzed the results of the PCR using fluorescent images (Figure 6A). The green fluorescence of the microchannel became much brighter due to dequenching of the FAM-labelled probe [39]. The normalized intensity of the microchannel increased more than sixfold after the PCR process as compared to before (Figure 6D), showing the successful amplification of target DNA. To measure the fluorescence intensity before and after the PCR process, the normalized fluorescence intensity was measured by Image J software. The average values before and after the PCR process were analyzed with three different experiments. They confirmed that the fluorescence intensity significantly increased with DNA amplification after PCR. The mechanical property of the bonded PMMA-based plastic lab-on-a-chip was analyzed by a tensile test (Figure 6B), indicating that Young’s modulus of the bonded PMMA-based plastic lab-on-a-chip was constant regardless of the PCR process. Additionally, there was no observable leakage of fluorescent dye molecules after PCR process and the normalized fluorescence intensity was also constant during all PCR cycles (Figure 6C). Therefore, we successfully demonstrated that our CuS/rGO-PEG nanocomposite-mediated photothermal bonding was a highly robust and promising method for biological applications of the PMMA-based plastic lab-on-a-chip.

## 4. Conclusions

We developed CuS/rGO-PEG nanocomposites for photothermal bonding in a PMMA-based plastic lab-on-a-chip. We optimized the volume of CuS/rGO-PEG nanocomposites for photothermal bonding and further analyzed the mechanical properties of the PMMA-based plastic lab-on-a-chip, showing that the tensile force was proportional to droplet number of CuS/rGO-PEG nanocomposites. Furthermore, the Young’s modulus and fluorescence intensity of the PMMA-based plastic lab-on-a-chip were measured over 35 thermal cycles for PCR applications, showing that there was no significant change due to the PCR process. Therefore, this noncontact photothermal bonding could be a potentially powerful method in various plastic labs-on-chip for pandemic molecular diagnostic applications.

## Figures and Tables

**Figure 1 nanomaterials-11-00176-f001:**
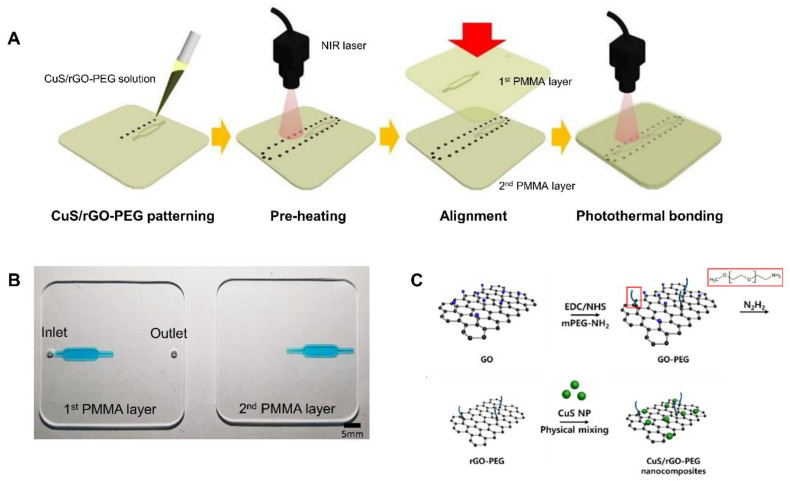
Polymethyl methacrylate (PMMA)-based plastic lab-on-a-chip using copper sulfide (CuS)/reduced graphene oxide (rGO)-poly(ethylene glycol) (PEG) nanocomposites: (**A**) schematic illustration of PMMA-based plastic lab-on-a-chip bonding using the photothermal effect, (**B**) photograph of the PMMA-based plastic lab-on-a-chip, (**C**) synthesis process of the CuS/rGO-PEG nanocomposite.

**Figure 2 nanomaterials-11-00176-f002:**
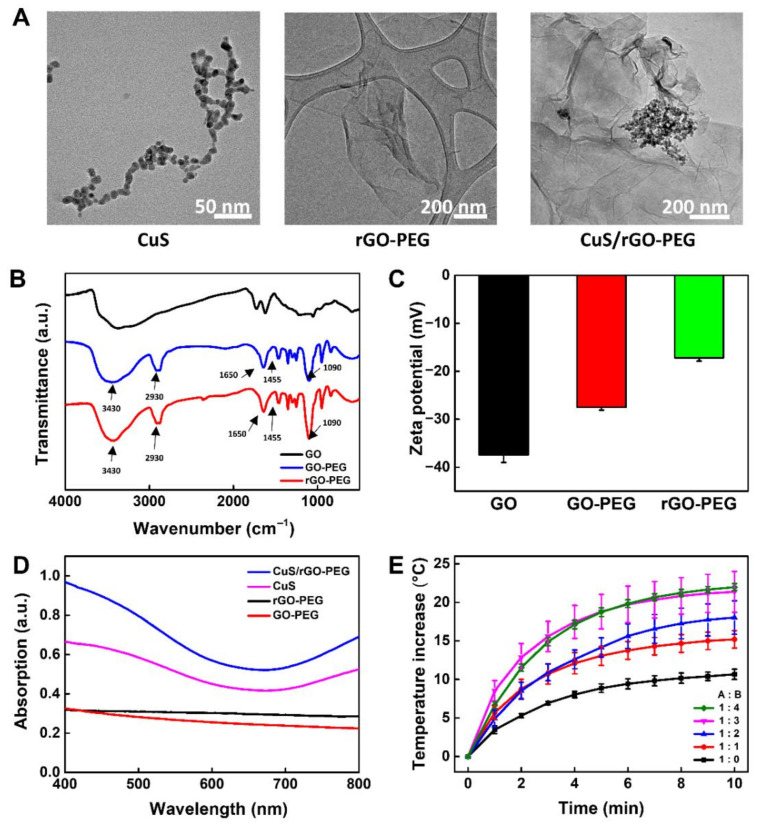
Characterization of the CuS/rGO-PEG nanocomposite: (**A**) TEM images of CuS nanoparticles, rGO-PEG, and CuS/rGO-PEG nanocomposites. (**B**) FT-IR analysis results for GO, GO-PEG, and rGO-PEG nanocomposites. (**C**) Zeta-potential analysis of GO, GO-PEG, and rGO-PEG nanocomposites. (**D**) UV-vis spectroscopy analysis of GO-PEG, rGO-PEG, CuS, and CuS/rGO-PEG nanocomposite. (**E**) Photothermal effect by the ratio of (**A**) rGO-PEG (0.25 mg/mL) with (**B**) CuS nanoparticle under 808 nm near-infrared (NIR) irradiation (1 W/cm^2^).

**Figure 3 nanomaterials-11-00176-f003:**
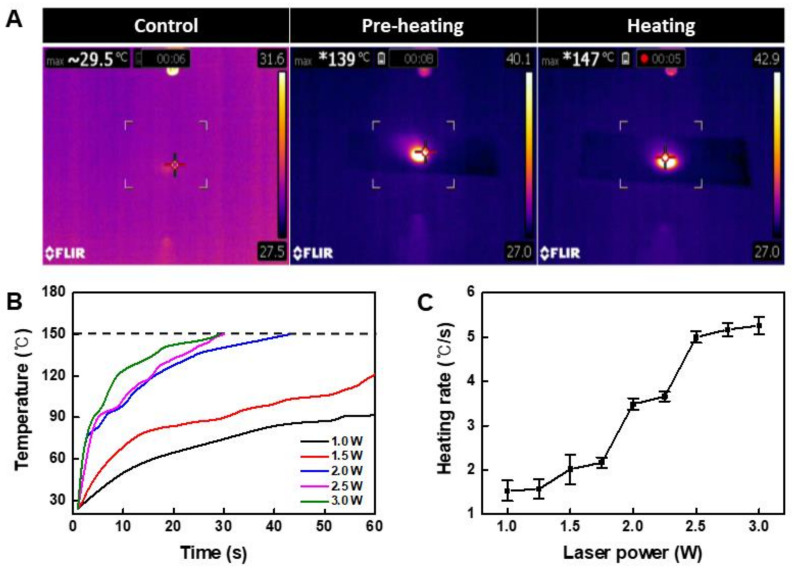
Temperature analysis of CuS/rGO-PEG nanocomposites: (**A**) infrared thermal images of CuS/rGO-PEG nanocomposites on the PMMA-based plastic lab-on-a-chip in the preheating and heating step, (**B**) temperature analysis of CuS/rGO-PEG nanocomposites with various laser intensities under 808 nm NIR irradiation, (**C**) heating rate of CuS/rGO-PEG nanocomposites with various laser intensities.

**Figure 4 nanomaterials-11-00176-f004:**
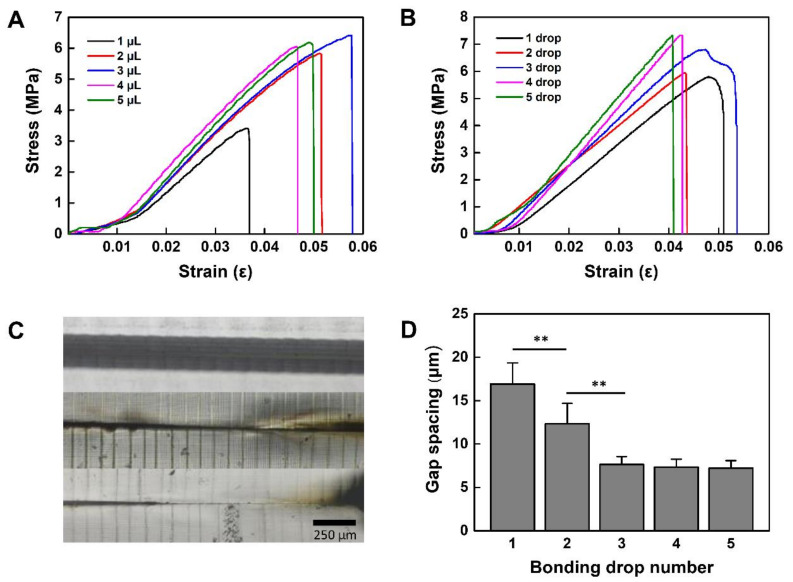
Mechanical properties of photothermally bonded PMMA-based plastic lab-on-a-chip: (**A**) tensile strength with various volumes of nanocomposite, (**B**) tensile strength with the number of drops, (**C**) gap of the bonded PMMA-based plastic lab-on-a-chip with the number of droplets, (**D**) length of gap spacing of bonded PMMA-based plastic lab-on-a-chip relative to bonding drop numbers (** *p* < 0.01).

**Figure 5 nanomaterials-11-00176-f005:**
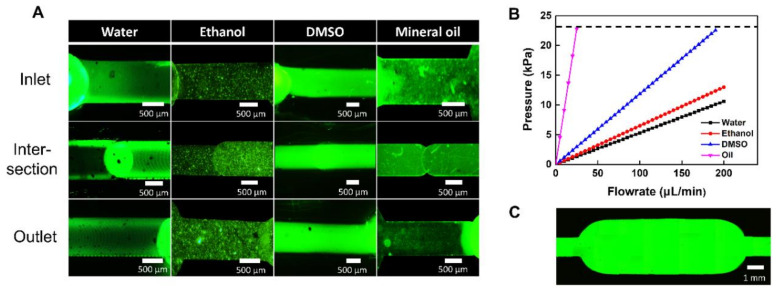
Leak test of PMMA-based plastic lab-on-a-chip after photothermal bonding: (**A**) fluorescent images of bonded PMMA-based plastic lab-on-a-chip leak test with fluids of varying viscosity (water, ethanol, DMSO, mineral oil), (**B**) persistence analysis of bonded PMMA-based plastic lab-on-a-chip, (**C**) fluorescent image of bonded PMMA-based plastic lab-on-a-chip without leaking.

**Figure 6 nanomaterials-11-00176-f006:**
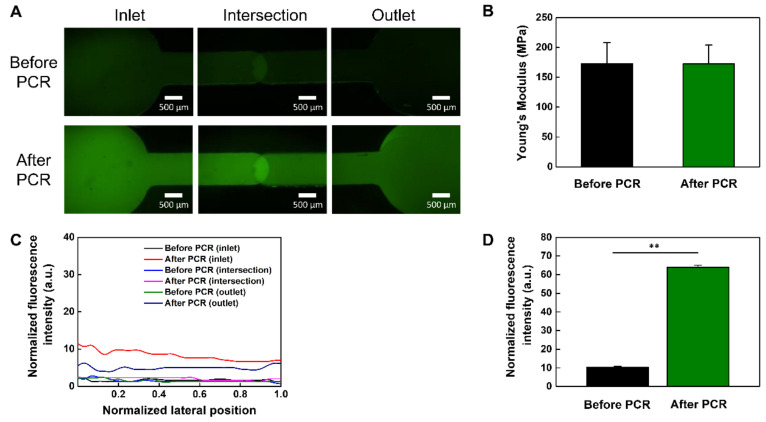
PCR application in the PMMA-based plastic lab-on-a-chip: (**A**) fluorescent images of before and after PCR in the PMMA-based plastic lab-on-a-chip, (**B**) Young’s modulus analysis before and after PCR, (**C**) normalized fluorescence intensity analysis with respect to the channel position of PMMA-based plastic lab-on-a-chip before and after the PCR, (**D**) normalized fluorescence intensity analysis before and after PCR in the PMMA-based plastic lab-on-a-chip (** *p* < 0.01).

## Data Availability

The data presented in this study are available on request from the corresponding author.

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
