# Peer review of "CuS/rGO-PEG Nanocomposites for Photothermal Bonding of PMMA-Based Plastic Lab-on-a-Chip"

_nanomaterials, 2021, doi:10.3390/nano11010176_

Round 1

Reviewer 1 Report

The manuscript "CuS/rGO-PEG nanocomposites for photothermal bonding of PMMA-based plastic Lab-on-a-Chips" describes an innovative method for the bonding process of PMMA chips. The use of the photothermal effect for bonding application is very interesting and well described in the paper. In the sfollowing some minor suggestions:

Introduction: The introduction is well structured and focused from the beginning on the fabrication point of view of polymeric chips, probably it could be introduced a starting period describing the importance and various field of application of lab-on-a-chip devices (i.e. see https://doi.org/10.1021/acsami.7b02633 and references therein). I would also like to have some information on the cost related to the implementation of the proposed technique. Authors could add some line of comments in comparison with conventional bonding systems. 

I would also suggest to improve the figure captions.

Figure 4D is not clear, maybe authors could add some labels or better explain the figure in the caption or in the text.

Scale bars in some cases are not clearly visible

Reviewer 2 Report

In this study, Kim et al. developed the CuS/rGO-PEG nanocomposites for photothermal bonding of the PMMA-based plastic Lab-on-a-Chips. They demonstrated that non-contact photothermal bonding of PMMA-based plastic Lab-on-a-Chips plays an important role in improving the stability and adhesion at a high temperature as well as minimizing the solution leakage from microchannels when connecting two devices. In addition, no solution leakage in PMMA-based plastic Lab-on-a-Chips was found during polymerase chain reaction (PCR) thermal cycles. Overall, the logic is reasonable, the manuscript was well prepared, and the experiments were carefully conducted. I can recommend this manuscript draft to be published in this journal after addressing the following questions:

  1. The numbers from chemical formulas should be checked carefully, all numbers should be lowercased.
  2. The photograph of the final assembled device should be provided.
  3. From TEM images, it seems that the CuS nanoparticles are not monodispersed. And they are also not distributed uniformly onto the rGO. If there is a good way to fabricate the nanocomposite?
  4. The TEM images of different ratios of CuS to rGO-PEG should be provided for better presentation?
  5. Figure 4D and 6D, the statistical analysis method and results should be added.
  6. Table S1 can be removed to the main text.
  7. The authors tested the effect of flow rate and solvent type on pressure. Did the authors analyze the effect of operation time on the endurability of the PMMA-based plastic device?
  8. Since ethanol is also one commonly used solvent, it would be more meaningful to add the ethanol data in Figure 5B.

Reviewer 3 Report

The authors present NP-composites for optothermal bonding. Bonding states a. problem for any polymer microfluidics and the current approach is appealing. Prior to publication, the comparison of the composite NIR method with at least a second well-established method concerning bonding and stability would increase the relevance of the study. (e.g. PMMA compatible biomedical UV glue or solvent bonding)

Language and tenses are off throughout the manuscript. 

Figure 3: B,C add repititions for at least 3 batches

"Figure. 4A" should be "Figure 4A (Page 7 Line 258)

Figure 4AB: The force values should be normalized against the adhesion area which is different for increasing volumes of composite

Figure 4CD: Why is 4 and 5 drops missing in the graph?

Figure 6: How is biocompatibility relevant for a PCR-setup? 

Round 2

Reviewer 2 Report

All my questions have been properly solved, I can recommend it to be published as it is.

Author Response

Thanks for accepting this paper.

Reviewer 3 Report

The authors revised the manuscript accordingly.

small remark:

- bicompatibility is only used for interaction of materials with living hosts. Pcr is not a living system and the authors should find a better terminology
